# ATF-4 and hydrogen sulfide signalling mediate longevity in response to inhibition of translation or mTORC1

Cyril Statzer[1,9], Jin Meng [2,3,4,9], Richard Venz[1], Monet Bland[2,3,4], Stacey Robida-Stubbs[2,3,4], Krina Patel[2,3,4], Dunja Petrovic[5], Raffaella Emsley[6], Pengpeng Liu[7], Ianessa Morantte[8], Cole Haynes [7], William B. Mair [8], Alban Longchamp [6], Milos R. Filipovic[5], T. Keith Blackwell [2,3,4,10✉] & Collin Y. Ewald [1,10✉]

Inhibition of the master growth regulator mTORC1 (mechanistic target of rapamycin complex 1) slows ageing across phyla, in part by reducing protein synthesis. Various stresses globally suppress protein synthesis through the integrated stress response (ISR), resulting in preferential translation of the transcription factor ATF-4. Here we show in *C. elegans* that inhibition of translation or mTORC1 increases ATF-4 expression, and that ATF-4 mediates longevity under these conditions independently of ISR signalling. ATF-4 promotes longevity by activating canonical anti-ageing mechanisms, but also by elevating expression of the transsulfuration enzyme CTH-2 to increase hydrogen sulfide ($H_2S$) production. This $H_2S$ boost increases protein persulfidation, a protective modification of redox-reactive cysteines. The ATF-4/CTH-2/$H_2S$ pathway also mediates longevity and increased stress resistance from mTORC1 suppression. Increasing $H_2S$ levels, or enhancing mechanisms that $H_2S$ influences through persulfidation, may represent promising strategies for mobilising therapeutic benefits of the ISR, translation suppression, or mTORC1 inhibition.

[1] Eidgenössische Technische Hochschule Zürich, Department of Health Sciences and Technology, Institute of Translational Medicine, Schwerzenbach, Switzerland. [2] Department of Genetics, Harvard Medical School, Boston, MA, USA. [3] Joslin Diabetes Center, Research Division, Boston, MA, USA. [4] Harvard Stem Cell Institute, Cambridge, MA, USA. [5] Leibniz-Institut für Analytische Wissenschaften-ISAS-e.V., Dortmund, Germany. [6] Department of Vascular Surgery, Centre Hospitalier Universitaire Vaudois and University of Lausanne, Lausanne, Switzerland. [7] Department of Molecular, Cell and Cancer Biology, University of Massachusetts Medical School, Worcester, MA, USA. [8] Department of Genetics and Complex Diseases, Harvard School of Public Health, 665 Huntington Avenue, Boston, MA, USA. [9] These authors contributed equally: Cyril Statzer, Jin Meng. [10] These authors jointly supervised this work: T. Keith Blackwell, Collin Y. Ewald. ✉email: keith.blackwell@joslin.harvard.edu; collin-ewald@ethz.ch

Over the last three decades, genetic and phenotypic analyses of ageing have revealed that across eukaryotes, lifespan can be extended by inhibition of mechanisms that promote growth and proliferation[1–3]. Prominent among these is the serine/threonine kinase complex mTORC1, which coordinates the activity of multiple growth-related processes in response to growth factor and nutrient signals[2–4]. mTORC1 activity can be reduced by genetic perturbations, dietary restriction (DR), or pharmacological interventions such as rapamycin, an mTORC1 inhibitor that increases lifespan from yeast to mice[2,3]. However, while rapamycin represents a very exciting paradigm for anti-ageing pharmacology, mTORC1 suppression has wide-ranging effects on the organism[2,3]. Rapamycin is used clinically as an immunosuppressant, and mTORC1 broadly affects metabolism and supports the synthesis of proteins, nucleic acids, and lipids[2–4]. Elucidation of specific mechanisms through which mTORC1 influences longevity is critical not only for understanding the biology of ageing and longevity, but also for the development of molecularly targeted anti-ageing therapies that maintain health.

mTORC1 increases the rates at which numerous different mRNAs are translated, and a hallmark of mTORC1 inhibition is a reduction in overall protein synthesis[2,3]. Work in the model organisms *C. elegans* and *Drosophila* indicates that lifespan extension from mTORC1 inhibition is mediated in part through this global decrease in translation[2,5,6]. Furthermore, in *C. elegans* suppression of translation is sufficient to increase both lifespan and stress resistance[7–12]. A mechanistic understanding of how mRNA translation levels affect longevity should therefore provide mechanistic insights into how mTORC1 influences lifespan.

Suppression of new protein synthesis is also an important mechanism through which cells protect themselves from stressful conditions that include nutrient deprivation, and thermal-, oxidative-, and endoplasmic reticulum (ER) stress[13,14]. Those stresses induce the evolutionarily conserved ISR by activating kinases that phosphorylate and inhibit the translation initiation factor subunit eIF2α, thereby imposing a broad reduction in cap-dependent mRNA translation[13,14]. This suppression of translation leads in turn to preferential translation of the activating transcription factor ATF4, which mobilizes stress defense mechanisms to re-establish homeostasis[13,14]. Although the ISR has important protective functions, its effects on longevity and health are complex. In *C. elegans*, genetic or pharmacological interventions that impair the ISR extend lifespan by inducing preferential translation of selective mRNAs[15]. In older mice, pharmacological ISR inhibition enhances memory and cognition by allowing protein synthesis to be maintained[14,16]. On the other hand, in *S. cerevisiae* the ATF4 ortholog Gcn4 promotes longevity[17,18], and in *C. elegans* hexosamine pathway activation enhances proteostasis through the ISR and ATF-4[19]. It remains to be determined whether ATF4 is required and sufficient to promote metazoan longevity.

Here we have investigated whether and how ATF-4 affects lifespan in *C. elegans*. We find that ATF-4 but not upstream ISR signalling is required for longevity induced by conditions that inhibit protein synthesis, and that ATF-4 is a pro-longevity factor that extends lifespan when overexpressed on its own. ATF-4 increases lifespan not only by enhancing canonical anti-ageing mechanisms, but also by inducing transsulfuration enzyme-mediated $H_2S$ production and therefore levels of protein persulfidation. Importantly, the anti-ageing benefits of mTORC1 suppression depend upon this ATF-4-induced increase in $H_2S$ production, further supporting the idea that they derive from lower translation rates and suggesting that increases in ATF-4 and $H_2S$ levels may recapitulate these benefits.

## Results

**ATF-4 responds to translation suppression to increase *C. elegans* lifespan.** We investigated whether *C. elegans* atf-4 is regulated similarly to mammalian ATF4 at the level of mRNA translation. In mammals, 2–3 small upstream open reading frames (uORFs) within the ATF4 5′ untranslated region (UTR) occupy the translation machinery under normal conditions, inhibiting translation of the downstream ATF4 coding region[14,20]. By contrast, when translation initiation is impaired by eIF2α phosphorylation, the last uORF is bypassed and ATF4 is translated preferentially[14,20]. The *C. elegans* atf-4 ortholog (previously named atf-5) contains two 5′ UTR uORFs (Fig. 1a, Supplementary Fig. 1a, b), deletion of which increases translation of a transgenic reporter[21], predicting that translation of the atf-4 mRNA may be increased under conditions of global translation suppression.

We tested this idea in *C. elegans* that express green fluorescent protein (GFP) driven by the atf-4 upstream region, including the two uORFs (P*atf-4*(uORF)::GFP, Fig. 1a). P*atf-4*(uORF)::GFP expression was extremely low under unstressed conditions, but increased when translation was suppressed by treatment with the translation elongation blocker cycloheximide, or RNA interference (RNAi)-mediated knockdown of the tRNA synthase rars-1 (Fig. 1b, Supplementary Fig. 1c–f). Compounds that induce ER stress and elicit the ISR, including tunicamycin (TM) or dithiothreitol (DTT), also strongly induced P*atf-4*(uORF)::GFP expression (Fig. 1b, Supplementary Fig. 1c, d). By contrast, TM treatment increased atf-4 mRNA levels by only 1.5-fold (Fig. 1c, Supplementary Fig. 1g). Treatment with alpha-amanitin, which blocks transcription, prevented TM from increasing the levels of the atf-4 mRNA but not P*atf-4*(uORF)::GFP fluorescence (Fig. 1c, d). Together, the data indicate that the P*atf-4*(uORF)::GFP reporter is regulated post-transcriptionally during ER stress. The endogenous atf-4 mRNA was expressed at steady levels during development and ageing (Supplementary Fig. 1h). By contrast, ribosome occupancy was decreased on its uORFs compared to the coding region as larval development progressed, with the lowest levels apparent during the L4 stage after the body plan has been formed and growth has slowed (Fig. 1e, f). This suggests that the endogenous atf-4 gene is regulated translationally through its uORF region under unstressed conditions, possibly in response to growth-related cues. We conclude that, like mammalian ATF4, *C. elegans* atf-4 is regulated translationally and preferentially translated upon conditions of reduced protein synthesis (Supplementary Fig. 1i).

Given that ATF-4 is upregulated by translation suppression during the ISR, we hypothesized that it might be important for lifespan extension by reduced protein synthesis. In *C. elegans*, RNAi-mediated knockdown of various translation initiation factors increases lifespan[7–12]. In the wild type (WT) background, neither a partial[21] nor a complete (Fig. 2a, b, Supplementary Data 1) ablation of atf-4 activity decreased lifespan. By contrast, the lifespan increases that occurred in response to knockdown of ifg-1/eIF4G, ife-2/eIF4E, or eif-1A/eIF1AY, were abrogated in atf-4(tm4397) loss-of-function mutants (Fig. 2a, Supplementary Data 1). Similarly, a low dose of cycloheximide extended the lifespan of WT but not atf-4(tm4397) animals (Fig. 2b, Supplementary Data 1). Thus, atf-4 is required for longevity arising from a global reduction in protein synthesis.

The increase in ATF-4 translation that occurs during the canonical ISR is induced by translation suppression that is imposed through increased eIF2α phosphorylation[22], but this might not be the case when translation is reduced by directly inhibiting translation initiation or elongation. Supporting this idea, the low dose of cycloheximide that was sufficient to extend lifespan (Fig. 2b) increased ATF-4 expression (Supplementary Fig. 1f), but not eIF2α phosphorylation (Fig. 2c, d). Similarly,

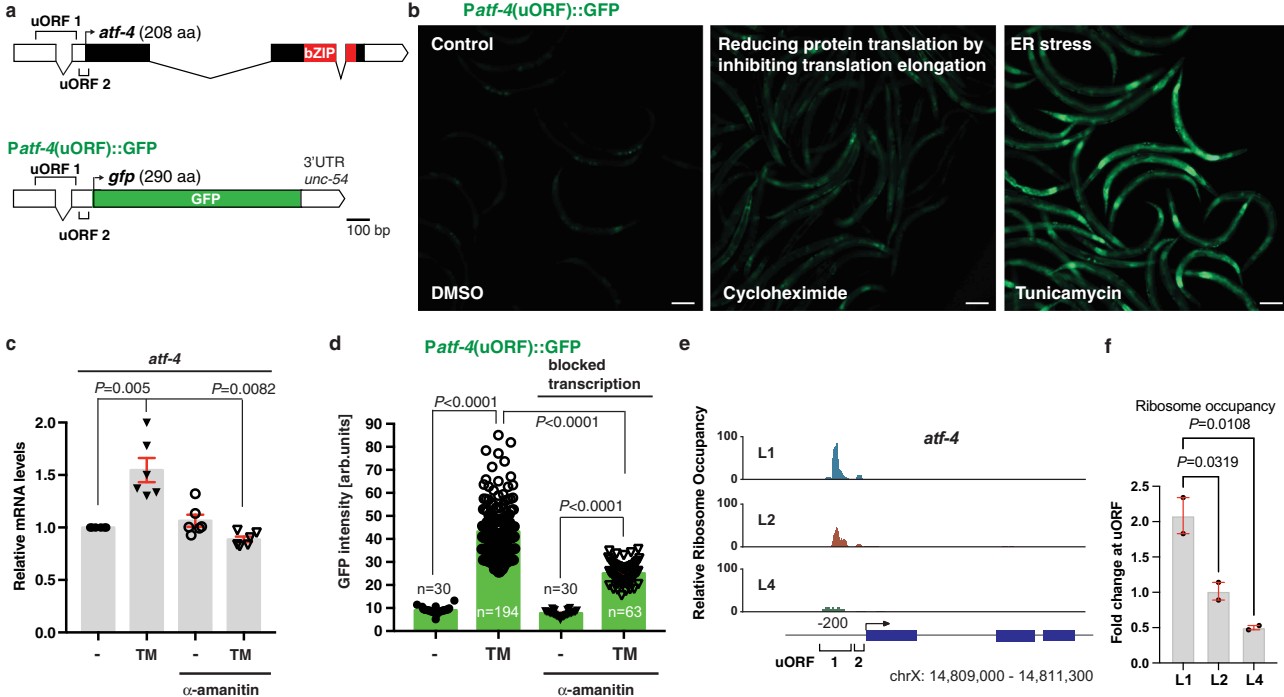

**Fig. 1 ATF-4 is preferentially translated under conditions of reduced global protein synthesis. a** Schematic diagram of the *atf-4* mRNA and the P*atf-4*(uORF)::GFP reporter. UTRs are represented as empty boxes, exons as filled boxes, and the basic leucine zipper domain (bZIP) in red. **b** Representative images showing that reducing translation by administering 7.2 mM cycloheximide for 1 hour or 35 μg/ml tunicamycin (TM) for 4 h increased expression of transgenic P*atf-4*(uORF)::GFP in L4 stage animals. Quantification of GFP fluorescence intensity is shown in Supplementary Fig. 1c. Scale bar = 100 μm. **c** A 1 h pre-treatment with 0.7 μg/ml α-amanitin (RNA Pol II inhibitor) prevented 4 h of 35 μg/ml TM treatment from increasing *atf-4* mRNA levels in L4 stage animals. Mean ± SEM. Three independent trials, measured in duplicates. *P* values are relative to untreated wild type (N2) determined by one sample *t*-test, two-tailed, hypothetical mean of 1. **d** A 1 h pre-treatment with 0.7 μg/ml α-amanitin did not prevent TM treatment from increasing levels of transgenic P*atf-4*(uORF)::GFP expression in L4 stage animals. Mean ± SEM. *n* > 30 animals examined over 2 independent experiments, One-way ANOVA with post hoc Tukey. **e** Stage-specific ribosome occupancy profiles of the endogenous *atf-4* mRNA, along with quantification of relative uORF occupancy (**f**). Analysis of ribosomal profiling data[69] revealed a decrease in ribosome occupancy on the endogenous *atf-4* uORFs under unstressed conditions during late larval development. Mean ± SD, *n* = 2 biological replicates. Occupancy profiles were generated by assigning counts to the *atf-4* transcript based on the number of raw reads at each position. Blue boxes indicate the *atf-4* exons. One-way ANOVA post hoc Dunnett's test.

depletion of the translation initiation factor *ifg-1*/eIF4G reduced protein synthesis and dramatically increased expression of P*atf-4*(uORF)::GFP, but knocking down either *ifg-1* or *eif-1A* only modestly increased eIF2α phosphorylation (Fig. 2e–i). This suggests that treatments that inhibit translation can increase ATF-4 expression without triggering canonical induction of the ISR through eIF2α phosphorylation.

We next investigated whether translation inhibition might increase lifespan independently of this canonical ISR signalling, using a well-characterized eIF2α mutant (*eif-2α(qd338)*) in which the serine at which inhibitory phosphorylation occurs during the ISR (S49 in *C. elegans*; S51 in mammals) is mutated to phenylalanine, so that eIF2α phosphorylation and ISR induction are blocked[23]. A mutation that prevents phosphorylation of this serine partially suppresses lifespan extension from reduced insulin/IGF-1 signalling, suggesting that ISR signalling is important for longevity in this context[24]. Importantly, the *eif-2α(qd338)* mutation did not interfere with the *atf-4*-dependent increase in lifespan that was seen with *ifg-1* knockdown (Fig. 2j). We conclude that canonical ISR signalling through eIF2α phosphorylation is not necessarily required for translation inhibition to induce preferential ATF-4 translation or increase lifespan through ATF-4.

**ATF-4 mobilises canonical pro-longevity mechanisms.** In *C. elegans*, a limited number of transcription factors have been identified that increase lifespan when overexpressed, including DAF-16/FOXO, HSF-1/HSF1, and SKN-1/NRF[1,25]. These evolutionarily conserved regulators are generally associated with enhancement of protective mechanisms such as stress resistance, protein folding or turnover, and immunity. To determine whether ATF-4 can actually promote longevity, as opposed to being required generally for health, we investigated whether an increase in ATF-4 levels might extend lifespan. Transgenic ATF-4-overexpressing animals (ATF-4OE) exhibited nuclear accumulation of ATF-4 in neuronal, hypodermal, and other somatic tissues under unstressed conditions (P*atf-4*::ATF-4(gDNA)::GFP; Supplementary Fig. 2a). TM treatment doubled ATF-4 protein levels (Supplementary Fig. 2b, Source data File), indicating that this ATF-4 transgene responds to environmental stimuli. Importantly, ATF-4 overexpression increased lifespan by 7–44% across >10 independent trials, which included two experiments without 5-Fluoro-2′deoxyuridine (FUdR) and analysis of two independent transgenic lines (Fig. 3a, Supplementary Data 1). ATF-4 over-expression also prolonged healthspan (Fig. 3b, Supplementary Fig. 2c, Supplementary Data 2). We conclude that the elevated activity of the ATF-4 transcriptional program is sufficient to extend lifespan and promote health.

To identify longevity-promoting mechanisms that are enhanced by ATF-4, we used RNA sequencing (RNA-seq) to compare gene expression profiles in *atf-4* loss-of-function or ATF-4OE animals compared to WT under non-stressed conditions (Supplementary Fig. 3a, b, Supplementary Data 3). Only a modest number of genes were detectably up- or downregulated by *atf-4* loss or overexpression, respectively (Fig. 3c, Supplementary

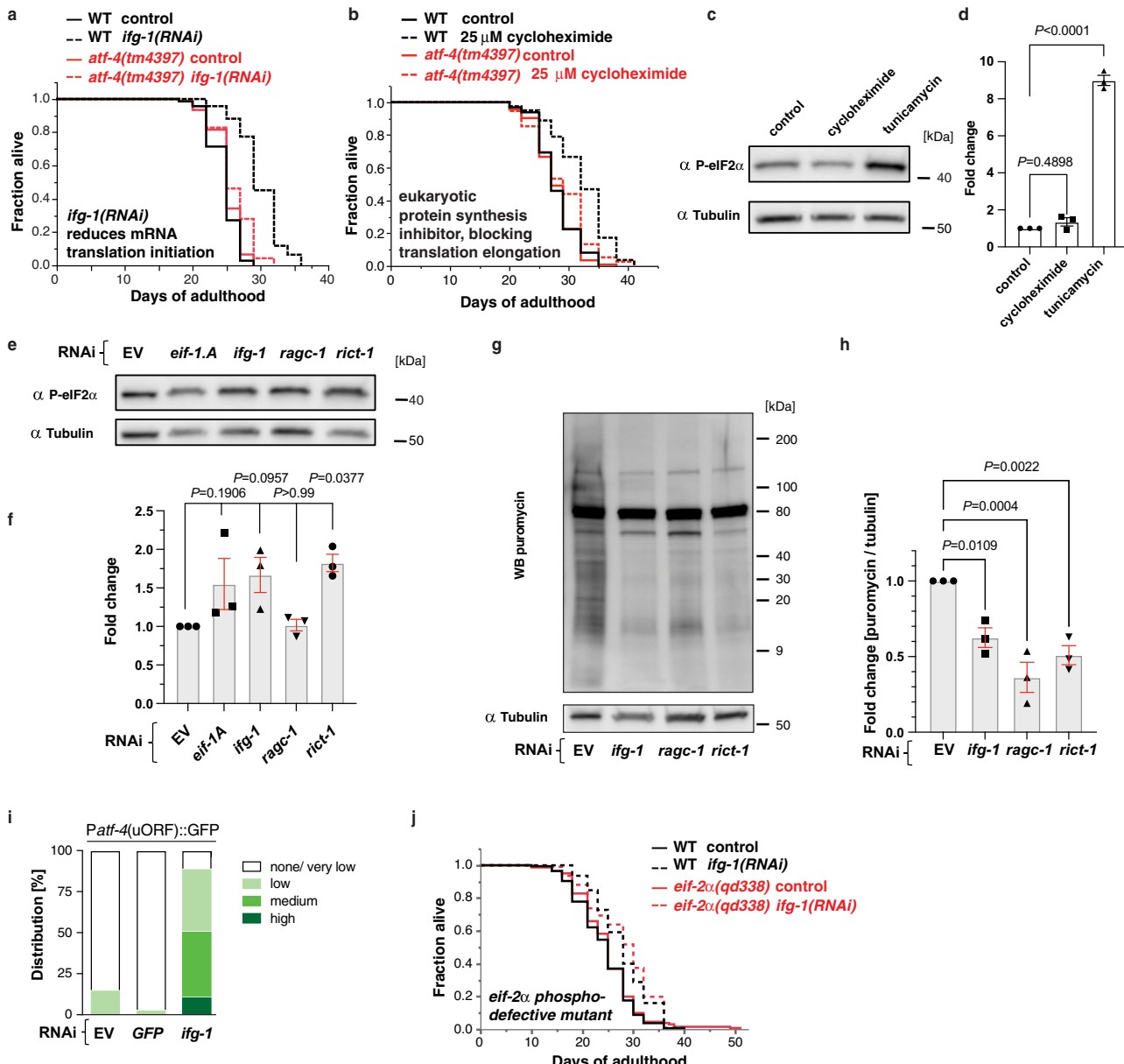

**Fig. 2 ATF-4 mediates lifespan extension from translation inhibition. a** Adult-specific knockdown of *ifg-1* extended the lifespan of *WT* animals but not *atf-4(tm4397)* mutants. **b** Adult-specific treatment with 25 μM cycloheximide increased lifespan dependent upon *atf-4*. **c** Representative western blots and quantification (**d**) showing that treatment with 35 μg/ml tunicamycin for 4 h dramatically increased eIF2α phosphorylation levels in L4 stage animals, while treatment with 7.2 mM cycloheximide for 1 h did not. Mean ± SEM. $n = 3$ biological replicates. One-way ANOVA with post hoc Tukey. **e** Representative western blots and quantification (**f**) showing the effects of adult-specific knockdown of *eif-1.A*, *ifg-1*, *ragc-1*, or *rict-1* on eIF2α phosphorylation levels. Mean ± SEM. $n = 3$ biological replicates. One-way ANOVA with Dunnett's post-test compared to EV. **g** Representative western blots of puromycin incorporation assay and quantification (**h**) showing that adult-specific knockdown of *ifg-1*, *ragc-1*, or *rict-1* decreased translation. Mean ± SEM. $n = 3$ biological replicates. One-way ANOVA with Bonferroni post-test. **i** Quantification of GFP fluorescence showing that adult-specific *ifg-1* knockdown increases expression of P*atf-4*(uORF)::GFP. **j** Adult-specific knockdown of *ifg-1* comparably extended the lifespan of *WT* animals and *eif-2α/y37e3.10(qd338)* phosphorylation-defective mutants. For statistics and additional trials in (**a**), (**b**), and (**j**), see Supplementary Data 1. For western blots, source data are provided in Source data File.

Fig. 3c, d). Notably, ATF-4 overexpression upregulated several small heat shock protein (HSP) genes that are also controlled by HSF-1/HSF (heat shock factor) and DAF-16/FOXO (Fig. 3d), and are typically induced by longevity-assurance pathways[1]. Each of the ATF-4-upregulated chaperone genes *sip-1*/CRYAA, *hsp-70*/HSPA1L, *hsp-16.2*/HSPB1, and *hsp-12.3*/HSPB2 was required for lifespan extension from ATF-4 overexpression (Supplementary Fig. 3e; Supplementary Data 1). Translation of *atf-4* was increased within minutes by a heat shock (Supplementary Fig. 3f),

suggesting that ATF-4 functions in tandem with HSF-1/HSF1. Together, the data suggest that ATF-4 enhances proteostasis mechanisms that have been linked to longevity.

Other findings further linked ATF-4 to longevity-associated mechanisms. ATF-4 overexpression increased expression of the cytoprotective gene *nit-1*/Nitrilase (Fig. 3d), a canonical target of the xenobiotic response regulator SKN-1/NRF[25], along with expression of collagen genes that are typically upregulated by SKN-1/NRF in response to lifespan extension interventions

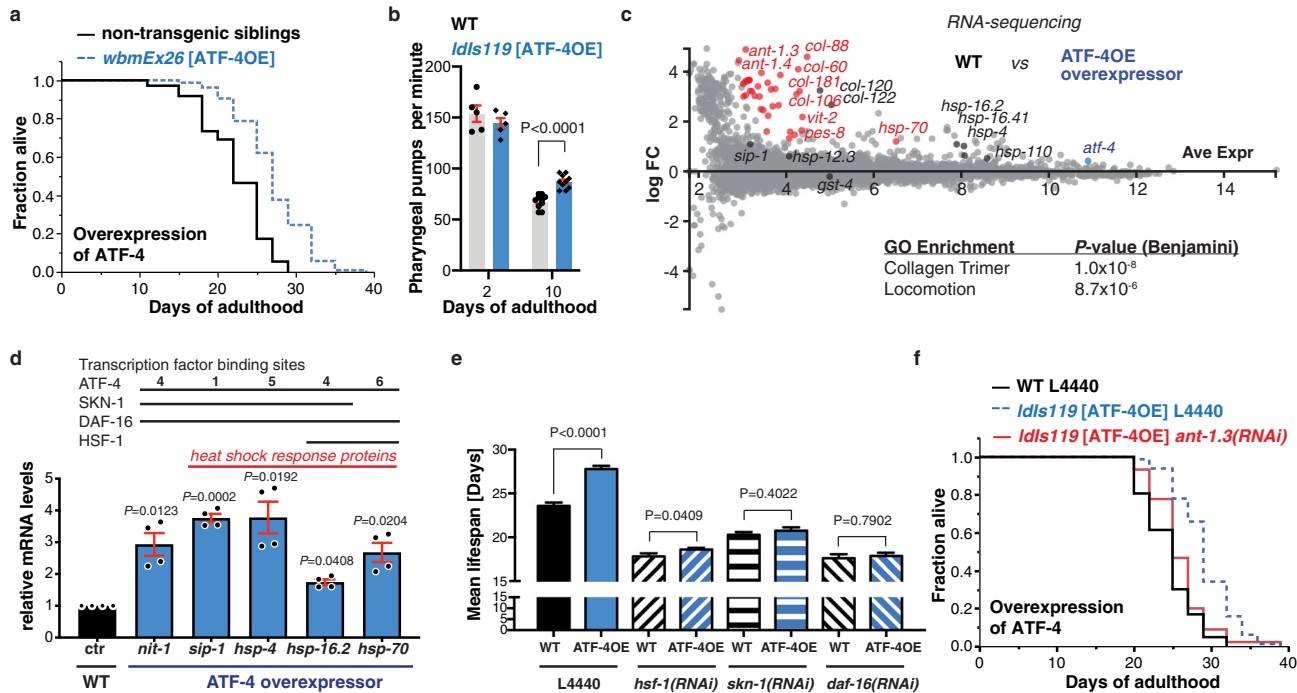

**Fig. 3 ATF-4 overexpression is sufficient to increase lifespan. a** Transgenic animals (*wbmEx26* [P*atf-4*::ATF-4(gDNA)::GFP]) that overexpress ATF-4 (ATF-4OE) live longer compared to their non-transgenic siblings. **b** Pharyngeal pumping rate is similar at day 2 of adulthood between ATF-4OE (*ldIs119* [P*atf-4*::ATF-4(gDNA)::GFP]) and WT, but higher in ATF-4OE at day 10 of adulthood, suggesting an improved healthspan. For the complete time-course of pharyngeal pumping rate during ageing, see Supplementary Data 2. Mean ± SEM. At least 5 worms were examined for each genotype and time point in one experiment. Unpaired two-tailed *t*-test. **c** MA (log ratio and mean average)-plot of RNA sequencing analysis comparing ATF-4OE(*ldIs119*) to WT, showing absolute log fold change (FC). In red, highlighted genes with FDR < 0.1 and log FC > 1 compared to WT. In black, genes with FDR > 0.1. Details are in Supplementary Data 3. **d** Validation by qRT-PCR of genes differentially expressed in ATF-4OE(*ldIs119*), using two new independent biological samples of about 200-215 animals each measured in duplicates. Mean ± SEM. *P* values relative to WT determined by one sample *t*-test, two-tailed, hypothetical mean of 1. The numbers of ATF4 binding sequences (-TGATG-)[27,28] are indicated in Supplementary Data 4. The DAF-16 and SKN-1 transcription factor binding sites are based on ChIP data from www.modencode.org (Supplementary Data 5). **e** Longevity conferred by ATF-4OE(*ldIs119*) is abolished by knockdown of *hsf-1*, *skn-1*, or *daf-16*. Mean ± SEM of Kaplan–Meier survival plot. *P*-value determined by log-rank. *n* = 1 biological replicate. **f** The mitochondrial ATP translocase *ant-1.3* is required for ATF-4 overexpression induced longevity. For statistical details and additional lifespan trials in (**a**), (**e**), and (**f**), see Supplementary Data 1.

(Fig. 3c)[26]. The 3 kb predicted promoter regions of many ATF-4-upregulated genes included not only the binding consensus for mammalian ATF4 (-TGATG-)[27,28], but also sites for DAF-16, HSF-1, and SKN-1 (Fig. 3d, Supplementary Data 4, 5). Furthermore, many genes that were upregulated by ATF-4 overexpression had been detected in chromatin immunoprecipitation (ChIP) analyses of these transcription factors (Supplementary Fig. 3g, Supplementary Data 5). Each of those transcription factors is critical for lifespan extension arising from suppression of translation[10,11], and we determined that they are also needed for longevity conferred by ATF-4 overexpression (Fig. 3e, Supplementary Data 1). ATF-4 overexpression also robustly upregulated two adenine nucleotide translocase genes (ANT; *ant-1.3* and *ant-1.4*, Fig. 3c). The ANT complex is important for transport of ATP from the mitochondrial space into the cytoplasm, as well as for mitophagy[29]. Both *ant-1.3* and *ant-1.4* were required for longevity by ATF-4 overexpression (Fig. 3f, Supplementary Data 1). Together, our findings suggest that while the transcriptional impact of ATF-4 may seem limited in breadth, it cooperates with other longevity factors to enhance the activity of multiple mechanisms that protect cellular functions, thereby driving lifespan extension.

**ATF-4 increases lifespan through H₂S production.** To identify ATF-4-regulated genes that are conserved across species and might be particularly likely to have corresponding roles in humans, we queried our ATF-4OE vs WT RNA-seq results, and compared the top 200 significantly upregulated *C. elegans* genes against 152 mammalian genes that are thought to be regulated directly by ATF4[28]. Seven orthologues of these genes were upregulated in ATF-4OE (Fig. 4a, Supplementary Data 4), four of which encoded components of the reverse transsulfuration (hereafter referred to as transsulfuration) pathway (*cth-2*/CTH), or associated mechanisms (*glt-1*/SLC1A2, *C02D5.4*/GSTO1, and *F22F7.7*/CHAC1) (Fig. 4b, Supplementary Data 4). The transsulfuration pathway provides a mechanism for utilising methionine to synthesize cysteine and glutathione when their levels are limiting[30], but the CTH enzyme (cystathionine gamma-lyase, also known as CGL or CSE) also generates H₂S as a direct product. Underscoring the potential importance of the H₂S-generating enzyme CTH-2 for ATF-4 function, the levels of its mRNA and protein were each increased by ATF-4 overexpression (Fig. 4c–e, Source data File).

Regulation of amino acid biosynthesis genes is a conserved ATF4 function[13], and reduced levels of methionine[31] and higher levels of H₂S[32,33] have each been linked to longevity. We did not detect any differences in the relative abundance of amino acids between ATF-4OE and WT animals (Supplementary Data 6), suggesting that ATF-4 is unlikely to influence longevity by altering amino acid levels. By contrast, ATF-4 overexpression consistently increased H₂S production capacity in a *cth-2-*

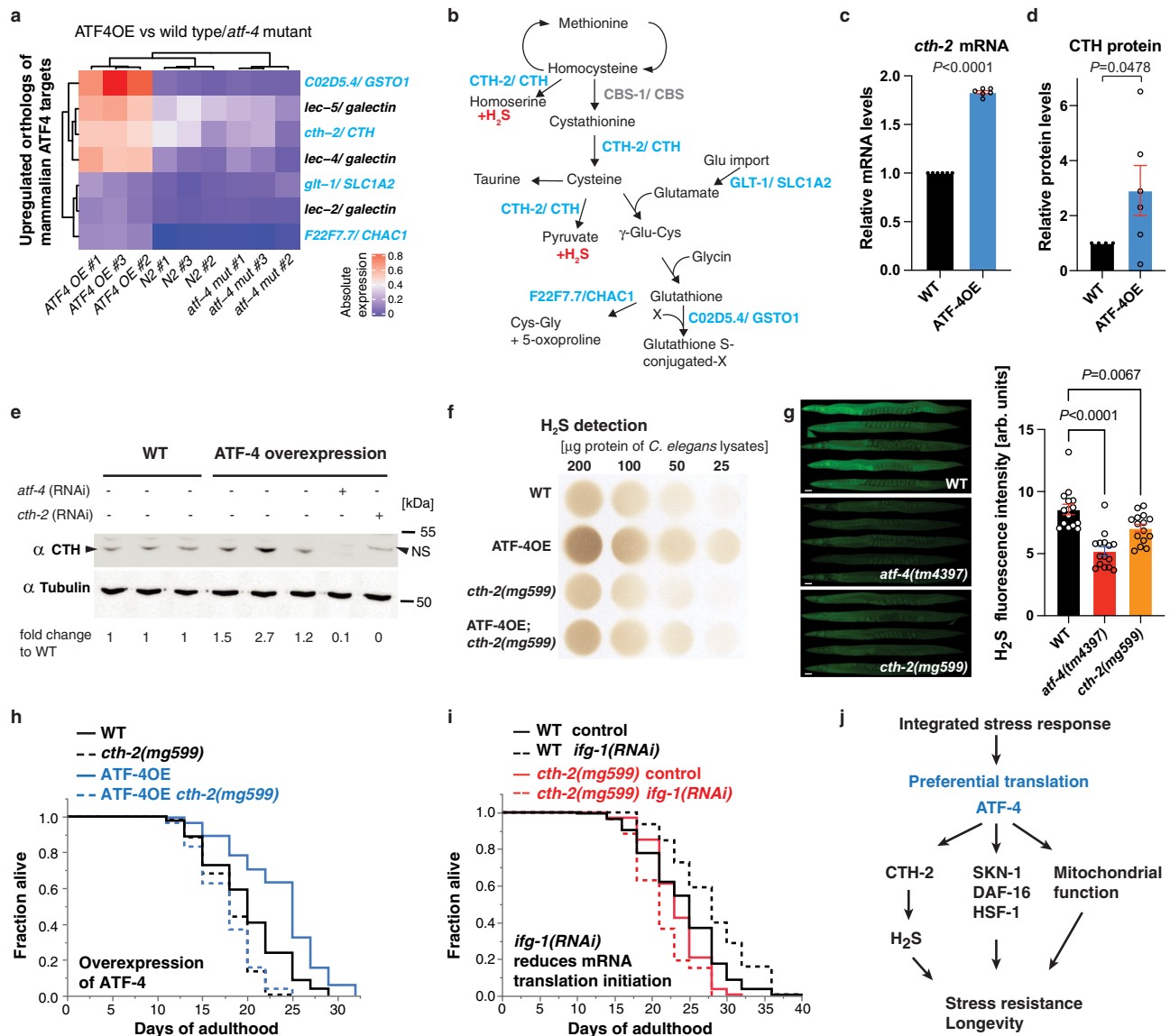

**Fig. 4 ATF-4 overexpression increases H₂S levels via cystathionine gamma-lyase, which is required for longevity. a** Heatmap of gene expression in ATF-4OE (*ldls119*), wild type (N2), and *atf-4(tm4397)* showing genes whose orthologs are directly regulated by mammalian ATF4 (Details are in 'Methods', Supplementary Data 4). Absolute levels of expression were compared. Genes in light blue are predicted to be involved in the transsulfuration pathway, which is shown in (**b**). **c** ATF-4OE(*ldls119*) showed higher *cth-2* mRNA levels compared to WT by qRT-PCR. $n = 3$ independent biological samples in duplicates (each over 200 L4 worms). Mean ± SEM. *P* values relative to WT determined by one-sample *t*-test, two-tailed, a hypothetical mean of 1. **d** Quantification of CTH protein levels in ATF-4OE(*ldls119*) compared to WT. $n = 6$ independent biological trials probed in 3 western blots. One-tailed *t*-test. **e** Western blots showing an ATF-4-induced increase in CTH levels was abolished by *atf-4* or *cth-2* knockdown. NS = non-specific band. **f** ATF-4 overexpression increases H₂S production capacity in a *cth-2*-dependent manner. Additional biological trials are shown in Supplementary Fig. 4d. For H₂S quantification, see Supplementary Data 12. **g** Representative fluorescent microscopy images and quantification showing that H₂S levels in vivo were decreased in either *atf-4(tm4397)* or *cth-2(mg599)* mutants compared to WT. Data are represented as Mean ± SEM. $n = 3$ biological replicates of a total of 15 worms per condition. *P* values to WT are unpaired *t*-test, two-tailed. Scale bar = 50 µm. **h** Lifespan extension induced by ATF-4 overexpression depends upon *cth-2*. **i** Lifespan extension induced by *ifg-1* knockdown requires *cth-2*. **j** Model for how ATF-4 promotes stress resistance and longevity. For statistical details and additional trials in (**h**) and (**i**), see Supplementary Data 7. For western blots, source data are provided in Source data File.

dependent manner (Fig. 4f, Supplementary Fig. 4a–d). Using the fluorescent H₂S probe MeRho-Az[34], we also found that H₂S levels are reduced by mutation of either *atf-4* or *cth-2* (Fig. 4g). Taken together, our data indicate that ATF-4 promotes H₂S production by acting through CTH-2. The increases in longevity and stress resistance that are conferred by ATF-4 overexpression were abrogated by *cth-2* mutation or knockdown (Fig. 4h, Supplementary Fig. 4e, Supplementary Data 1, 7). Similarly, *ifg-1*/eIF4G knockdown failed to extend lifespan in *cth-2* mutant animals (Fig. 4i). We conclude that the increase in H₂S production that

derived from CTH-2 upregulation is a critical and beneficial aspect of ATF-4 function (Fig. 4j).

An important consequence of increased H₂S levels is an increase in the proportion of protein cysteine thiols (-SH) that are converted to persulfides (-SSH)[35,36]. Redox modification at reactive cysteine residues is critical in growth signalling and other mechanisms[36], and in *C. elegans* thousands of redox-regulated cysteine residues are present in proteins that are involved in translation regulation, lipid and carbohydrate metabolism, stress signalling, and other fundamental biological

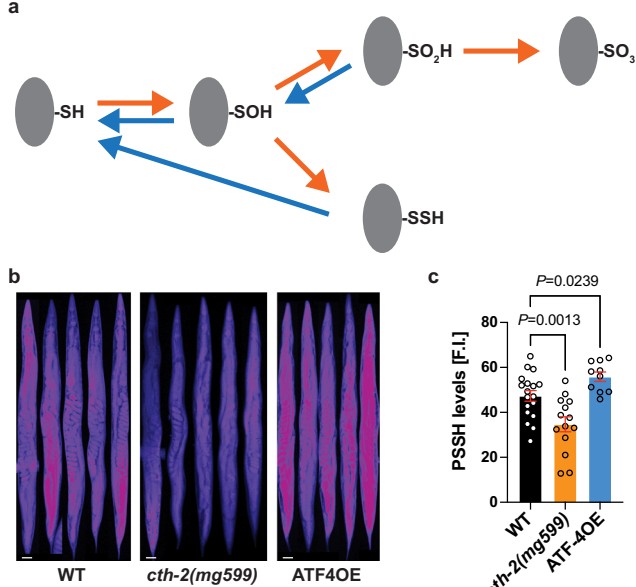

**Fig. 5 ATF-4 and CTH-2 regulate protein persulfidation levels. a**
Schematic diagram showing that the thiol group (–SH) of reactive cysteine residues in proteins can undertake various redox states. Sulfenylation (–SOH) can be reversed, particularly efficiently through the intermediate of persulfidation (–SSH), but sulfinylation (–SO₂H) is reversible only within peroxiredoxins and sulfonylation (–SO₃H) is irreversible[35,37]. Arrows in orange indicate oxidation processes while those in blue indicate reduction processes. **b** Representative fluorescent images and quantification (**c**) showing that ATF-4OE exhibited higher persulfidation levels, while *cth-2(mg599)* animals exhibited lower global persulfidation levels, compared to WT. Data are represented as Mean ± SEM. *n* = 3 biological replicates of at least 10 worms per condition. Scale bar = 50 μm. F.I. = fluorescent intensity. *P* values to WT are unpaired *t*-test, two-tailed.

processes[37]. Under oxidising conditions, these thiols are readily converted to sulfenic acids (–SOH), which is a reversible and in many cases regulatory modification that can proceed to irreversible and potentially damaging oxidized forms (–SO₂H, –SO₃H) (Fig. 5a)[36]. H₂S converts –SOH to –SSH in the process called persulfidation, preventing protein overoxidation and thereby preserving protein functions under stressed conditions (Fig. 5a)[35,36]. The levels of overall protein persulfidation (PSSH) can be visualised in individual animals by using chemical probes and confocal microscopy[35]. In *C. elegans* PSSH levels are decreased by mutation of the *cth-2* paralog *cth-1*, suggesting that they depend upon a background level of H₂S produced by the latter[35]. We found that PSSH levels are also reduced by mutation of either *cth-2* or *atf-4*, and are modestly increased by ATF-4 overexpression (Fig. 5b, c). Taken together, our results show that ATF-4 acts through multiple mechanisms to promote stress resistance and longevity, and that a CTH-2-driven increase in H₂S production and persulfidation is an essential aspect of this program.

**A partial role for ATF-4 in some lifespan extension programs.** Given that *atf-4* is required for lifespan extension in response to reduced translation, we investigated whether *atf-4* and its transcriptional target *cth-2* might be generally required for *C. elegans* longevity. Although ATF4/ATF-4 has been implicated in responses to mitochondrial stress or protein synthesis imbalance[27,38], *atf-4* was dispensable for the increases in lifespan or oxidative stress resistance that follow from developmental impairment of mitochondrial function (Supplementary Fig. 5a–c,

Supplementary Data 1, 8). The extent of lifespan extension by reduced insulin/IGF-1 signalling or germ cell proliferation was decreased by *atf-4* mutation but did not depend upon *cth-2*, consistent with other transsulfuration components and H₂S producers being implicated in the latter pathway (Supplementary Fig. 5d, e, Supplementary Data 1)[39]. The ATF-4-CTH-2 pathway of H₂S production may therefore be fully essential specifically when lifespan extension is driven by suppression of protein synthesis.

We also investigated whether the ATF-4-CTH-2 pathway might be involved in longevity induced by DR, which extends lifespan in essentially all eukaryotes. An increase in H₂S production capacity has been implicated in mediating some DR benefits in mammals[33]. In *C. elegans*, *atf-4* was not required for lifespan to be extended by a liquid culture food-dilution DR protocol but was partially required for lifespan extension in the genetic DR-related model *eat-2(ad1116)* (Supplementary Fig. 5f, Supplementary Data 1). Consistent with these findings, transsulfuration pathway genes other than *cth-2* are also partially required for *eat-2* lifespan extension[33,35], suggesting that in *C. elegans* other pathways than ATF-4-CTH-2 might also increase H₂S production during DR. In mammals, restriction of sulfur-containing amino acids (methionine and cysteine) acts through ATF4 and CTH to boost endothelial H₂S levels and angiogenesis[40], and multiple longevity interventions increase CTH mRNA levels[41], suggesting a possible role for the ATF4-CTH pathway in DR. Supporting this idea, our bioinformatic analysis revealed that CTH mRNA levels were increased in various mouse tissues in response to DR (32/36 profiles), rapamycin (4/6 profiles), or growth hormone insufficiency (8/8 profiles) (Supplementary Fig. 5g, Supplementary Data 11). Therefore, ATF4-induced H₂S upregulation is likely to be evolutionarily conserved as a contributor to lifespan extension.

**Longevity from mTORC1 suppression is driven by ATF-4 and H₂S.** Because mTORC1 inhibition increases lifespan in part by reducing protein synthesis[2,5,6], we hypothesized that ATF-4 might be activated and required for lifespan extension when mTORC1 is inhibited. The heterodimeric RAG GTPases transduce amino acid signals to activate mTORC1 signalling, and are composed of RAGA-1 and RAGC-1 in *C. elegans*. mTORC1 is required for *C. elegans* larval development[2], but lifespan can be increased by RNAi-mediated knockdown of either *raga-1* or *ragc-1* during adulthood, or by a partial loss-of-function mutation of *raga-1*[2,3]. The former strategy allows mTORC1 activity to be reduced without any associated developmental effects. Adulthood RAG gene knockdown reduced protein synthesis (Fig. 2g, h), consistent with previous studies of mTORC1[2,3], but did not induce eIF2α phosphorylation (Fig. 2e, f). Under these conditions P*atf-4(uORF)*::GFP expression was increased robustly even when eIF2α phosphorylation was blocked genetically (Fig. 6a, b, Supplementary Data 9). We conclude that ATF-4 is preferentially translated when mTORC1 activity is reduced and translation rates are low, and that this occurs independently of the canonical ISR mechanism of increased eIF2α phosphorylation.

Importantly, the increases in lifespan extension, stress tolerance, and healthspan that resulted from loss of either *raga-1* or *ragc-1* function required *atf-4* but not phosphorylation of eIF2α (Fig. 6c–f, Supplementary Figs. 2c, 5c, 6a, Supplementary Data 1–2, 7, 8, 10), consistent with our analyses of P*atf-4(uORF)*::GFP activation. In a single trial, the longevity induced by *raga-1* knockdown was blunted by ATF-4 overexpression, suggesting that too much ATF-4 activity might be harmful (Supplementary Fig. 2c), but this has been observed for some other pro-longevity factors[25]. In summary, the data demonstrate

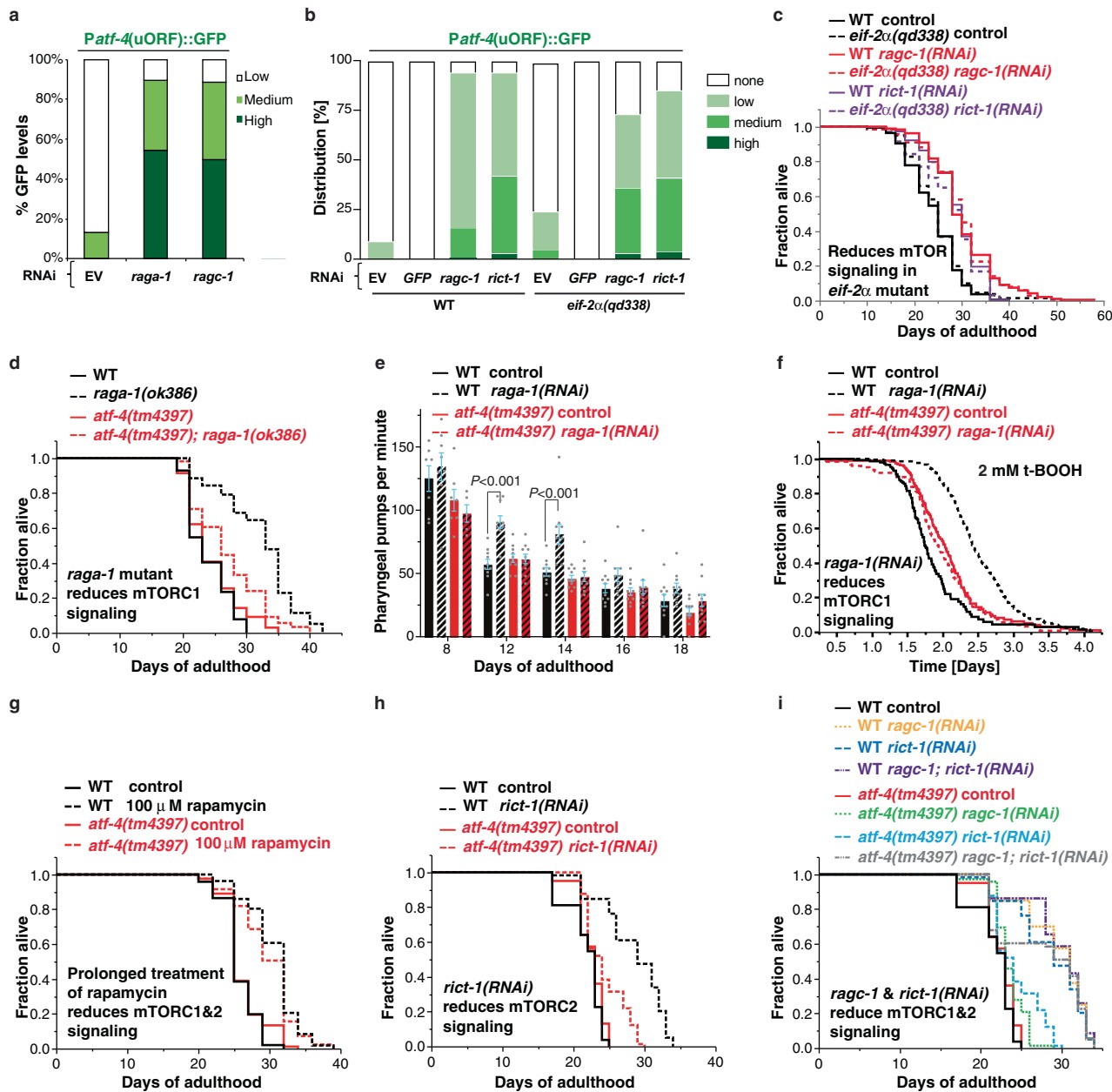

**Fig. 6 ATF-4 is essential for longevity from reduced mTORC1 activity. a** Inhibition of mTORC1 by either *raga-1* or *ragc-1* knockdown led to preferential translation of ATF-4. RNAi treatments were initiated at the L4 stage, with GFP intensity scored at day 3 of adulthood. **b** Inhibition of mTORC1 or mTORC2 by knockdown of *ragc-1* or *rict-1*, respectively, leads to preferential translation of ATF-4. Similar effects were observed in WT and *eif-2α(qd338)* mutants. **c** Post-development knockdown of *raga-1* or *rict-1* extends lifespan in both WT and *eif-2α(qd338)* mutants. **d** Mutation in *raga-1* increases lifespan in an *atf-4*-dependent manner. **e** Reducing mTORC1 signalling by adulthood specific *raga-1* knockdown improves healthspan dependent upon *atf-4*, as assessed by pharyngeal pumping rate. Mean ± S.E.M. *P* values relative to WT of the corresponding day with One-way ANOVA with post hoc Dunnett's multiple comparisons test. **f** Adult-specific knockdown *raga-1* increases oxidative stress resistance (2 mM tert-butyl hydrogen peroxide (tBOOH)) in an *atf-4*-dependent manner. RNAi was started at the L4 stage, and stress resistance was measured at day 3 of adulthood with the lifespan machine (See Supplementary Data 10 for details). **g** Rapamycin treatment during adulthood extends lifespan independently of *atf-4*. **h** Adult-specific knockdown of the mTORC2 subunit *rict-1* extends lifespan in an *atf-4*-dependent manner. **i** Adult-specific inactivation of both mTORC1 and mTORC2 increases lifespan independently of *atf-4*. For statistical details and additional trials in (**a**), (**b**), see Supplementary Data 9. For statistical details and additional lifespan trials in (**c**), (**d**), (**g**)-(**i**), see Supplementary Data 1.

that ATF-4 activation plays an essential role in the benefits of reducing mTORC1 activity.

Having determined that *atf-4* is required for mTORC1 suppression to extend lifespan, we were surprised to find that *atf-4* was dispensable for lifespan extension from rapamycin treatment, even though rapamycin increased ATF-4 translational reporter expression (Fig. 6g, Supplementary Fig. 6b, Supplementary Data 1, 9). Notably, mTOR is present in both the mTORC1

and mTORC2 complexes (Supplementary Fig. 6c)[3]. mTORC2 is not as well understood as mTORC1, but it functions in growth signalling and its activation involves binding to the ribosome, suggesting an association with translation regulation (Supplementary Fig. 6c)[42]. Rapamycin mechanistically inhibits mTORC1 activity, but continuous rapamycin treatment also reduces mTORC2 activity by blocking complex assembly[43,44], leading us to investigate the possible involvement of *atf-4* in mTORC2

effects. Knockdown of the essential mTORC2 subunit RICT-1 (Rictor) suppressed translation (Fig. 2g, h) and increased P*atf-4*(uORF)::GFP expression independently of eIF2α phosphorylation (Figs. 2e, f, 6b, Supplementary Data 9). The effects of mTORC2 on *C. elegans* lifespan are complex, but adulthood RNAi knockdown of *rict-1* extends lifespan[2,6]. Lifespan extension by *rict-1* knockdown required *atf-4* (Fig. 6h, Supplementary Data 1). Interestingly, however, simultaneous inactivation of mTORC1 and mTORC2 by knocking down both *raga-1* and *rict-1* extended lifespan independently of *atf-4* (Fig. 6i, Supplementary Data 1), as occurred with rapamycin treatment, suggesting that when the activity of both mTOR complexes is suppressed the requirement for *atf-4* is relieved.

We investigated whether an ATF-4-mediated increase in H$_2$S production is required for lifespan extension arising from inhibition of mTORC1 or mTORC2. Genetic inhibition of either mTORC1 or mTORC2 increased H$_2$S production capability in an *atf-4*-dependent manner (Fig. 7a, b, Supplementary Fig. 6d, e). Furthermore, the ATF-4 target gene *cth-2* was required for the increased lifespan of animals with reduced mTORC1 or mTORC2 activity (Fig. 7c, d, Supplementary Data 1, 7), suggesting that an ATF-4/CTH-2-mediated increase in H$_2$S production is essential for the benefits of either mTORC1 or mTORC2 inhibition. Knockdown of *cth-2* prevented mTORC1 inhibition from increasing stress resistance, further supporting this idea (Extended data Fig. 6f). Interestingly, simultaneous knockdown of *raga-1* and *rict-1* dramatically increased H$_2$S production capability, and this was greatly reduced but not eliminated in the *atf-4* loss-of-function mutant (Supplementary Fig. 6g). This result, together with the dispensable role for ATF-4 in longevity under these conditions, suggests that simultaneous inhibition of both mTOR complexes mobilizes mechanisms that increase lifespan independently of *atf-4* and possibly H$_2$S.

The observation that PSSH levels are reduced by lack of either ATF-4 or CTH-2 (Figs. 5b, c, 7e, f) suggests that the ATF-4/CTH-2 pathway might increase persulfidation in response to interventions that boost H$_2$S production through this pathway, including mTORC1 inhibition. Accordingly, in *raga-1* mutant adults PSSH levels were elevated in an *atf-4*-dependent manner (Fig. 7e, f). Taken together, our results show that reduced mTORC1 signalling leads to preferential translation of ATF-4, which acts through CTH-2 to promote stress resilience and healthy ageing by increasing H$_2$S production, and possibly through the resulting increase in PSSH levels across the proteome (Fig. 7g).

## Discussion

We have identified ATF-4, the transcriptional effector of the ISR, as a pro-longevity factor that can extend *C. elegans* lifespan when overexpressed. Furthermore, conditions that reduce mRNA translation, including mTORC1 inhibition, increase ATF-4 expression without activating canonical ISR signalling that is downstream of eIF2α phosphorylation, and require ATF-4 for lifespan extension. Previous studies revealed that longevity arising from inhibition of translation depends upon preferential translation of protective genes[45] and increased transcription of stress defence genes[6,10]. Our findings link these mechanisms by revealing that preferentially translated ATF-4 cooperates with DAF-16/FOXO, HSF-1/HSF, and SKN-1/NRF to drive protective gene transcription. Viewed alongside the well-documented pro-longevity activity of *S. cerevisiae* Gcn4 (ATF-4 ortholog)[17,18] and evidence that ATF-4 levels and activity are increased in long-lived mouse models[46,47], our results indicate that ATF-4 has an ancient and broadly conserved function in promoting longevity.

A striking aspect of our findings is that the longevity and health benefits of ATF-4 depend upon activation of its target transsulfuration pathway gene *cth-2*, and the resultant increase in H$_2$S production. For many years an understanding of how H$_2$S might promote health and longevity proved to be elusive[48], but recent work has implicated increased PSSH and its salutary effects on the proteome[35]. This modification protects the proteome from the effects of oxidative stress by "rescuing" sulfenylated cysteine residues from the fate of further oxidation (Fig. 5a), declines during ageing, and is increased in other long-lived models[35]. Here we demonstrated that PSSH levels were increased or decreased by ATF-4 overexpression or *cth-2* mutation, respectively, and that mTORC1 inhibition increased PSSH levels through ATF-4 (Figs. 5, 7). These results provide the first definition of a regulatory pathway through which a pro-longevity intervention increases H$_2$S and PSSH levels (Fig. 7g). Although persulfides can be introduced during translation[40], our results align with previous evidence that H$_2$S levels may largely determine the extent of this protective cysteine modification[35]. It was particularly intriguing that the ATF-4/CTH-2 pathway, which boosts H$_2$S formation and PSSH levels, was required for lifespan extension from mTORC1 suppression. This ATF-4/H$_2$S-induced posttranslational shift in PSSH levels could influence many biological functions, including the activity of redox-regulated signalling pathways[35,37], making it of interest to elucidate how these modifications mediate the downstream effects of mTORC1 signalling.

Our data add to the evidence that translation suppression is an essential effector of the longevity effects of mTORC1 inhibition[5,6]. We were surprised, however, to find that mTORC2 knockdown decreased protein synthesis levels and depended upon ATF-4 for lifespan extension (Figs. 2, 6). It will be very interesting to elucidate the mechanisms underlying the former observation. It was also surprising that either simultaneous mTORC1 and mTORC2 inhibition or rapamycin extended lifespan independently of ATF-4, suggesting that when both mTOR complexes are inhibited an independent mechanism compensates for lack of ATF-4. An understanding of how this occurs is likely to identify additional mechanisms that promote longevity.

Our evidence that reduced mTORC1 activity promotes longevity by increasing ATF-4 levels contrasts with mammalian evidence that pharmacological mTORC1 inhibition reduces ATF4 translation[20,41]. However, those findings were obtained in cultured cells in which mTORC1 activity was elevated genetically or by growth factor treatment, a very different scenario from adult *C. elegans* in vivo, in which growth has largely ceased and most tissues are post-mitotic. It seems logical that mTORC1 might increase ATF4 translation under the former conditions, given the importance of mTORC1 for translation of many genes and the need to maintain amino acid levels under conditions of high growth activity. On the other hand, it is consistent with our *C. elegans* results that analyses of mouse liver found that ATF4 protein levels and activity are increased in long-lived models that include rapamycin treatment and nutrient restriction[46,47], and that mTORC1 hyperactivation (TSC1 deletion) decreases CTH expression and prevents DR from increasing *CTH* mRNA levels[33]. It will be interesting in the future to determine how mammalian mTORC1 influences ATF4 in vivo under different conditions, including analysis of tissues with varying rates of growth and levels of mTORC1 activity.

Although inhibition of mTORC1 has received widespread enthusiasm as an anti-ageing strategy, mTORC1 maintains fundamental processes that include protein synthesis, mRNA splicing, and metabolic pathways[2,3,41,49,50], suggesting that not all effects of mTORC1 suppression are necessarily beneficial. Similarly, although conditions that suppress ISR signalling can

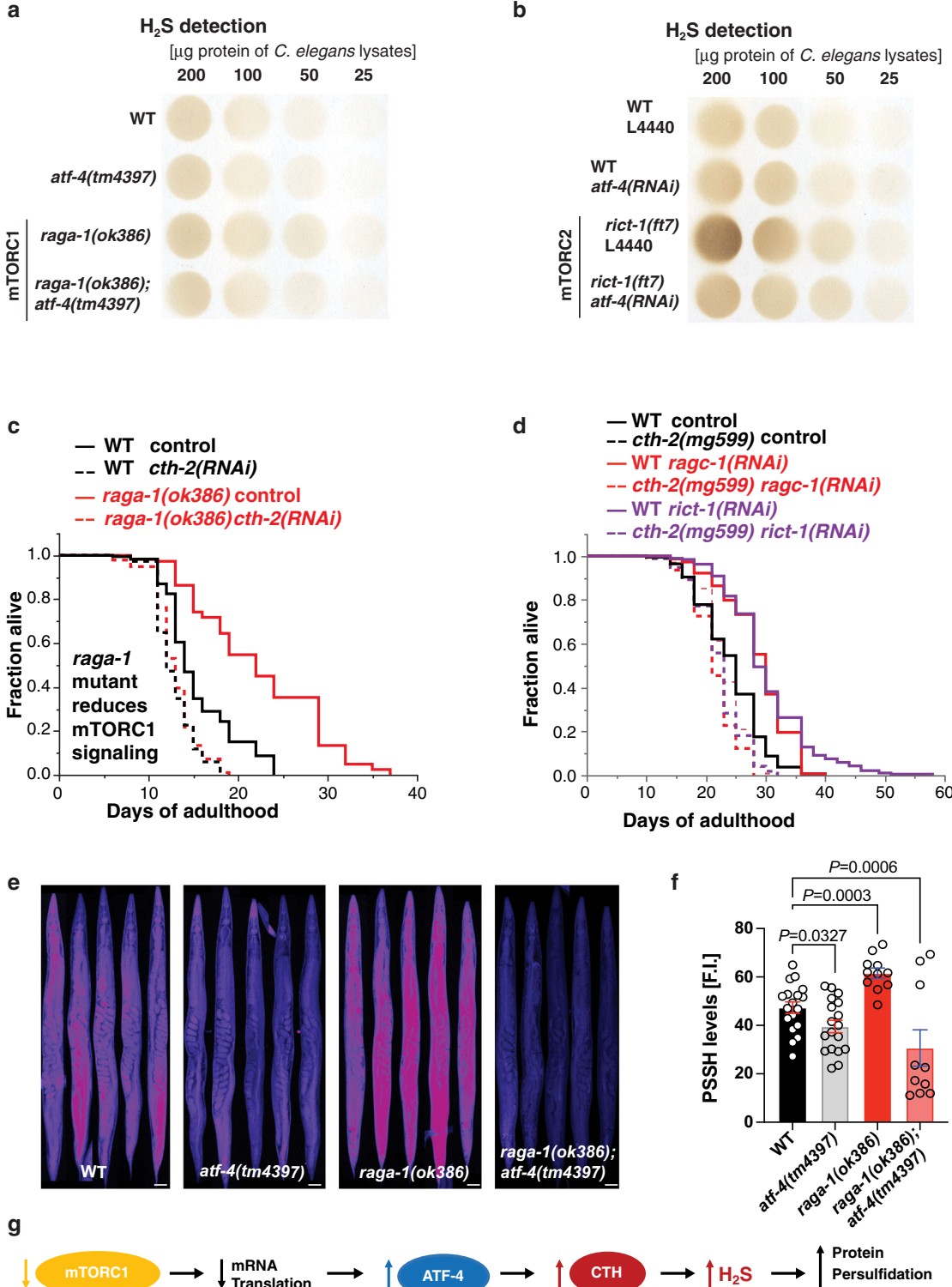

**Fig. 7 Longevity from mTOR inhibition upregulates H₂S and requires *cth-2*. a** Assay of *C. elegans* lysates showing that *raga-1* mutation increased H₂S production capacity in an *atf-4*-dependent manner. Two additional independent biological trials are shown in Supplementary Fig. 6d. **b** Assay of *C. elegans* lysates showing that *rict-1* mutation increased H₂S production capacity in an *atf-4*-dependent manner. An additional independent biological trial is in Supplementary Fig. 6e. **c** Longevity of *raga-1(ok386)* mutants is ablated by *cth-2* knockdown. This particular experiment was performed at 25 °C. **d** Longevity induced by adult-specific knockdown of either *raga-1* or *rict-1* depends upon *cth-2*. **e, f** Representative images showing persulfidation levels in WT (N2), *cth-2 (mg599)*, *atf-4 (tm4397)*, *raga-1 (ok386)*, and *raga-1;atf-4* mutants. Data are represented as Mean ± SEM. *n* = 3 biological replicates of at least 10 worms per condition. Scale bar = 50 μm. *P* values to WT are unpaired *t*-test, two-tailed. **g** Inhibition of mTORC1 promotes longevity by increasing ATF-4 expression and stimulating H₂S production. For H₂S quantification in (**a**, **b**), see Supplementary Data 12. For statistical details and additional lifespan trials in (**c**, **d**), see Supplementary Data 1.

promote longevity through effects on selective translation[15], ISR signalling through eIF2α contributes to lifespan extension from reduced insulin/IGF-1 signalling[24], and our results demonstrate that the downstream ISR effector ATF-4 is a potent pro-longevity factor. Furthermore, while pharmacological ISR inhibition preserves cognitive functions during ageing by maintaining protein synthesis[14,16], our findings suggest that ISR suppression could reduce levels of $H_2S$, which has been shown to prevent neurodegeneration[51,52]. In these and other settings, targeted mobilisation of beneficial mechanisms that are activated by ATF-4, including $H_2S$ production, might be of promising long-term value. Consistent with this notion, $H_2S$ confers many cardiovascular benefits in mammals, including a reduction in blood pressure[52,53], and patients suffering from vascular diseases show reduced CTH and $H_2S$ levels[54], prompting clinical trials of $H_2S$-releasing agents for cardiovascular conditions (NCT02899364 and NCT02278276). It could be of considerable value to examine the potential benefits of ATF4 and $H_2S$ in various settings, including the prevention of ageing-related phenotypes and disease.

## Methods

**Strains**. *Caenorhabditis elegans* strains were maintained on NGM plates and OP50 *Escherichia coli* bacteria. LD1499 [P*atf-4*(uORF)::GFP::*unc-54*(3'UTR)] was made by Chi Yun (1.8 kb promoter 5' of *atf-4* including both uORFs into pPD95.75)[55]. All strains used are listed in Supplementary Table 1. The *C. elegans* work described here was not covered by regulations concerning the ethical treatment of animals and did not require approval by a committee.

**Generation of transgenic lines**. To construct a translational fusion of ATF-4 with GFP, the plasmid pWM48 (P*atf-4*::ATF-4(gDNA)::GFP::*unc-54*(3'UTR)) was generated by introducing the 1.8 kb promoter region 5' of *atf-4* and the *atf-4* genomic sequence into pAD1. This construct was used to generate two independent transgenic lines: *wbmEx26* [pWM48 (P*atf-4*::ATF-4(gDNA)::GFP::*unc-54*(3'UTR), pRF4 (*rol-6*(su1006))] and *wbmEx27* [pWM48 (P*atf-4*::ATF-4(gDNA)::GFP::*unc-54*(3'UTR), pRF4 (*rol-6*(su1006))]]. UV irradiation was used for integration resulting in *ldIs119* from *wbmEx26* and *ldIs120-1* from *wbmEx27*, which were outcrossed 8-10x against N2.

**Genomic organisation and alignments**. Alignment of *C. elegans* ATF-4 (T04C10.4, WBGene00000221, 208 amino acids, www.wormbase.org) with human ATF4 (350 amino acids, P18848; www.uniprot.org) or ATF5 (282 amino acids, Q9Y2D1; www.uniprot.org) sequences was performed by T-COFFEE (Version_11.00.d625267). The *atf-4* genomic representation was made using Exon-Intron Graphic Maker (http://wormweb.org/exonintron) from Nikhil Bhatla. DNA and mRNA sequences were from www.wormbase.org (WS258). For human *ATF4* GenBank BC008090 mRNA sequence was used. The uORFs were predicted with ApE- A plasmid Editor v2.0.50b3. For amino acid alignments T-COFFEE (Version_11.00.d625267) was used.

**Ribosome profiling analysis**. Ribosome profiling sequencing data were downloaded from the NCBI Sequence Read Archive (S.R.A.) (http://www.ncbi.nlm.nih.gov/sra/) under accession number SRA055804. Data were analyzed as described[54]: Data analysis was performed with the help of Unix-based software tools. First, the quality of raw sequencing reads was determined by FastQC (Andrews, S. FastQC (Babraham Bioinformatics, 2010)). Reads were then filtered according to quality via FASTQ for a mean PHRED quality score above 30 (http://usegalaxy.org/u/dan/p/fastq). Filtered reads were mapped to the worm reference genome (Wormbase WS275) using B.W.A. (version 0.7.5), and S.A.M. files were converted into B.A.M. files by SAMtools (version 0.1.19). Read counts were then assigned to each gene or non-coding RNA. Library sizes were normalized using the EdgeR software package and the TMM normalization mode. Ribosome profiling was computed separately for each library as the average normalized read counts across all for normalized libraries. Thus, the mapped reads were determined and normalized based on library size for each transcript in each ribosome profiling library. The ATF-4 coverage data for each larval stage (L1, L2 and L4) and for the whole transcript (including 5'UTR, exons and 3'UTR) were calculated and exported by SAMtools. The stage-specific averaged coverage data for each gene were plotted using R (https://www.r-project.org).

**Knockdown by RNA interference**. RNAi clones were obtained from the Vidal and Ahringer RNAi libraries. RNAi bacteria cultures were grown overnight in LB with carbenicillin [100 µg/ml] and tetracycline [12.5 µg/ml], diluted to an OD600 of 1, and induced with 1 mM IPTG and spread onto NGM plates containing tetracycline

[12.5 µg/ml] and ampicillin [50 µg/ml]. For empty RNAi vector (EV) plasmid pL4440 was used as control.

**Manual lifespan assays**. Adult lifespan was determined either with or without FUdR as described in Ewald and colleagues[56]. About 100 L4 *C. elegans* per strain were picked onto NGM plates containing OP50 bacteria. The next day, *C. elegans* (day-1-adults) were transferred onto either NGM plates containing 400 µM FUdR and OP50 bacteria or RNAi bacteria. For cycloheximide-treatment lifespan, day-1-adults were transferred on NGM OP50 plates either containing the solvent 0.25% dimethyl sulfoxide (DMSO) alone as a control or cycloheximide (Sigma #C7698) dissolved in 0.25% DMSO. The rapamycin lifespan and liquid DR lifespan assays were performed as described[6,56]. Animals were classified as dead if they failed to respond to prodding. Exploded, bagged, burrowed, or animals that left the agar were excluded from the statistics. The estimates of survival functions were calculated using the product-limit (Kaplan–Meier) method. The log-rank (Mantel–Cox) method was used to test the null hypothesis and calculate $P$ values (JMP software v.9.0.2.).

**Pharyngeal pumping**. Pharyngeal pumping was assessed as described previously[26]. Pharyngeal pumping was determined by counting grinder movements in 45 second intervals while the animals were feeding on the bacterial lawn.

**Puromycin assay**. Puromycin incorporation and detection assays were adapted from previous studies[15,57]. Approximately 500 L4 animals were resuspended in M9 and transferred to NGM plates containing 50 µM FUdR seeded with RNAi bacteria clones. After 3 days, worms were collected in M9 and then transferred to S-basal medium. Worms were incubated with 4 ml S-Basal that contained OP50 and 0.5 mg/ml puromycin for 1 h. Afterwards, worms were washed with S-basal for three times. Protein extraction and Western blots for puromycin detection were performed as described below.

**Quantitative real-time polymerase chain reaction (qRT-PCR) assays**. RNA was isolated with Trizol (TRI REAGENT Sigma), DNAse-treated, and cleaned over a column (RNA Clean & Concentrator™ ZYMO Research). First-strand cDNA was synthesized in duplicate from each sample (Invitrogen SuperScript III). SYBR green was used to perform qRT-PCR (ABI 7900). Primers are listed in Supplementary Table 2. For each primer set, a standard curve from genomic DNA accompanied the duplicate cDNA samples. mRNA levels relative to WT control were determined by normalizing to the number of *C. elegans* and the geometric mean of three reference genes (*cdc-42*, *pmp-3*, and Y45F10D.4). At least two independent biological replicates were examined for each sample. For statistical analysis, one-sample *t*-test, two-tailed, a hypothetical mean of 1 was used for comparison using Prism 6.0 software (GraphPad).

**RNA sequencing**. Three independent biological replicates were prepared by using sodium hypochlorite to harvest eggs and overnight L1 arrest in M9 buffer with 10 µg/ml cholesterol to synchronize *C. elegans*. For each sample, about 20000 *C. elegans* per strain were allowed to develop to the L4 stage under normal growth conditions on NGM OP50 plates at 20 °C (about 1000 *C. elegans* per one 10 cm NGM OP50 plate). WT, *atf-4(tm4397)*, and *ldIs119* were grown at the same time for each biological replicate. *C. elegans* were washed from the culturing NGM plates and washed additional 3 times with M9 buffer to wash away the OP50 bacteria. RNA was isolated with Trizol (TRI REAGENT Sigma), DNAse-treated, and cleaned over a column (RNA Clean & Concentrator™ ZYMO Research). The RNA was sent to Dana-Farber Cancer Institute Center core for sequencing (http://mbcf.dfci.harvard.edu). The RNA Integrity Number (RIN) was then assessed by using the Bioanalyzer 2100 (Agilent Technologies), and only samples with a high RIN score were used to prepare cDNA libraries. All nine samples were multiplexed in a single lane. Single read 50 bp RNA-sequencing with poly(A) enrichment was performed using a HiSeq 2000 (Illumina). We aligned the FASTQ output files to the *C. elegans* WBcel235 reference genome using STAR 2.4.0j software (http://code.google.com/p/rna-star/) with an average >80% coverage mapping the reads to the genome. The differential gene expression analysis was performed using Bioconductor (http://bioconductor.org) as described in[58]. Rsubread 1.16.1 feature-Counts was used to quantify the mapped reads in the aligned SAM output files. Transcripts with <1 count per million reads were discarded. Counts were scaled to Reads Per Kilobase of transcript per Million mapped reads (RPKM) and deposited as a final output file in (Supplementary Data 3). To analyze the differential expressed genes, we compared *atf-4(tm4397)*, and *ldIs119* to wild type using Degust (http://degust.erc.monash.edu) with the following settings: RPKM with minimum 5 counts using edgeR with a false discovery rate (FDR) of 0.1 and an absolute log fold change (FC) of 1 relative to WT. Results are displayed in MA-plots. Functional annotation clustering was performed with DAVID using high classification stringencies (https://david.ncifcrf.gov).

**Comparison of RNA sequencing data with mammalian ATF4 target genes**. The RNA-sequencing data described in the previous section was subjected to differential expression analysis using the limma package (Smyth, Gordon K.

"Limma: linear models for microarray data." Bioinformatics and computational biology solutions using R and Bioconductor. Springer, New York, NY, 2005. 397-420) available in the programming language R (Team, R. Core. "R: A language and environment for statistical computing." (2013): 201). The 200 most-upregulated genes that were identified by comparison of ATF4 OE to WT and passed a Benjamini–Hochberg adjusted P-value threshold of 0.1 were analyzed further. Mammalian ATF4-specific gene targets were obtained from Quiros et al.[27] and subjected to Ortholist2 to infer C. elegans orthologs based on a comparative genomic meta-analysis[59]. The intersection of the most-upregulated genes in our ATF4OE to WT expression analysis and the orthologs of the mammalian ATF4 targets is depicted as a heatmap showing all biological replicates (#1-3) (http://www.bioconductor.org/packages/devel/bioc/html/ComplexHeatmap.html). The atf-4 mutant samples are shown separately since the displayed genes were selected based on the comparison between ATF4OE and WT. The absolute expression levels are displayed in a blue (low) to white (medium) to red (high) colour gradient, with genes indicated as gene names or sequence names if the former is not available. Hierarchical clustering was applied to both genes (rows) and samples (columns). Additional information: GO term enrichment yielded a significant ($P = 0.047$, Benjamini–Hochberg corrected) enrichment of the membrane raft compartment (lec-2, lec-4, lec-5) while no significant enrichment for GO biological process, GO molecular function, KEGG- or REACTOME pathways were found.

**Analysis of CTH expression levels in mice.** Publicly-available expression datasets were analyzed to quantify the change of CTH expression levels in long-lived compared to normal-lived mice. A selected subset of comparisons displaying CTH upregulation in longevity is depicted in Fig. 6b, while the full table is provided in Supplementary Data 11. Microarray datasets and platform information were obtained from GEO (https://www.ncbi.nlm.nih.gov/geo/) followed by mapping probes to their corresponding genes and sequencing information was obtained from SRA (https://www.ncbi.nlm.nih.gov/sra) and processed using Trim Galore (https://www.bioinformatics.babraham.ac.uk/projects/trim_galore/) and Salmon[60]. Datasets were centered and scaled, and subsequently, the mean fold change, as well as its standard error, were computed for the CTH gene.

**Manual thermotolerance assays.** Day-1-adults were placed on NGM OP50 plates (maximum 20 C. elegans per plate) and placed in an incubator at 35 °C. Survival was scored every hour. Animals were classified as dead if they failed to respond to prodding. Exploded animals or animals that moved up on the side of the plate were censored from the analysis. The estimates of survival functions were calculated using the product-limit (Kaplan–Meier) method. The log-rank (Mantel–Cox) method was used to test the null hypothesis and calculate P values (JMP software v.9.0.2.).

**Automated survival assays using the lifespan machine.** Automated survival analysis was conducted using the lifespan machine described by Stroustrup and colleagues[61]. Approximately 500 L4 animals were resuspended in M9 and transferred to NGM plates containing 50 μM FUdR seeded with OP50 bacteria, RNAi bacteria supplemented with 100 μg/ml carbenicillin, heat-killed OP50 bacteria, or UV-inactivated E. coli strain NEC937 B (OP50 ΔuvrA; KanR) containing 100 μg/ml carbenicillin. For oxidative stress assays, tBOOH was added to 2 mM to the NGM immediately before pouring and seeding with heat-killed OP50 bacteria. Animals were kept at 20 °C until measurement. Heat and oxidative stress experiments were performed using regular petri dishes sealed with parafilm, while tight-fitting petri dishes (BD Falcon Petri Dishes, 50x9mm) were used for lifespan experiments. Tight-fitting plates were dried without lids in a laminar flow hood for 40 minutes before starting the experiment. Air-cooled Epson V800 scanners were utilized for all experiments operating at a scanning frequency of one scan per 10–30 min. Temperature probes (Thermoworks, Utah, U.S.) were used to monitor the temperature on the scanner flatbed and maintain 20 °C constantly. Animals which left the imaging area during the experiment were censored.

Population survival was determined using the statistical software R[62] with the survival and survminer (https://rpkgs.datanovia.com/survminer/) packages. Lifespans were calculated from the L4 stage (=day 0). For stress survival assays the moment of exposure was utilized to define the time point zero of each experiment.

**Manual oxidative stress assay (arsenite and tBOOH).** The manual oxidative stress assays were performed as described in detail in the bio-protocol[63]. L4 worms were manually picked onto fresh OP50 plates. The next day, 10–12 day-one old C. elegans were transferred into 24-well plates containing 1 mL M9 Buffer in quad-ruplicates for each strain and condition (three wells with sodium arsenite (Sigma-Aldrich) and one well M9 as control). For tBOOH stress assay, about 80 L4 C. elegans per condition were picked onto fresh RNAi plates. Three days later, 20 day-three-old C. elegans were picked onto NGM plates containing 15.4 mM tBOOH (Sigma-Aldrich). The survival was scored every hour until all animas died. Exploded animals were excluded from the statistics. The log-rank (Mantel–Cox) method was used to test the null hypothesis and calculate P values (JMP software v.9.0.2.).

**Oxidative stress assay by quantifying movement.** C. elegans were collected from NGM plates and washed four times by centrifugation, aspirating the supernatant and resuspending in fresh M9 buffer again. After the final wash, the supernatant was removed, and 10 μl of the C. elegans suspension pipetted into each well of a round-bottom 96-well microplate resulting in ~40–70 animals per well. To prevent desiccation, the wells were filled up immediately with either 30 μl M9, or 30 μl M9 containing 6.7 mM or 18.7 mM sodium arsenite yielding a final arsenite concentration of 0, 5, or 14 mM, respectively. Per C. elegans strain and conditions, we loaded two wells with M9 as control and six wells with either 5 or 14 mM arsenite as technical replicates. The plate was closed, sealed with Parafilm and briefly stirred and then loaded into the wMicrotracker device (NemaMetrix). Data acquisition was performed for 50 h, according to the manufacturer's instructions. The acquired movement dataset was analyzed using the dplyr (https://dplyr.tidyverse.org/reference/dplyr-package.html, version 1.0.7) and ggplot2 (https://ggplot2.tidyverse.org, version 3.3.5) R packages.

**H2S production capacity assay.** The H2S production capacity assay was adapted from Hine and colleagues[33]. Synchronized C. elegans at the L1 stage were placed on RNAi or OP50 food, and harvested from NGM plates as young adults if not specified otherwise. Afterwards, worms were washed four times by centrifugation and resuspension with M9 to remove residual bacteria. Approximately 3000 animals were collected as a pellet and mixed with the same volume of 2× passive lysis buffer (Promega, E194A) on ice. Three freeze-thaw cycles were performed by freezing the samples in liquid nitrogen and thawing them again using a heat block set to 37 °C. Particles were removed by centrifuging at 12,000 × g for 10 min at 4 °C. The pellet was discarded, and the supernatant used further. The protein content of each sample was determined (BCA protein assay, Thermo scientific, 23225) and the sample sequentially diluted with distilled water to the required protein mass range, usually 25–200 μg protein. To produce the lead acetate paper, we submerged chromatography paper (Whatman paper 3M (GE Healthcare, 3030-917)) in a 20 mM lead acetate (Lead (II) acetate trihydrate (Sigma, 215902-25G)) solution for one minute and then let it dry overnight. The fuel mix was prepared freshly by mixing Pyridoxal 5′-phosphate hydrate (Sigma, P9255-5G) and L-Cysteine (Sigma, C7352-25G) in Phosphate Buffered Saline on ice at final concentrations of 2.5 mM and 25 mM, respectively. A 96-well plate was placed on ice, 80 μl of each sample were loaded into each well and mixed with 20 μl fuel mix and subsequently covered using the lead acetate paper. The assay plate was then incubated at 37 °C for 3 h under a weight of approximately 1 kg to keep the lead acetate paper firmly in place. For analysis, the exposed lead acetate paper was imaged using a photo scanner. H2S levels were quantified as the amount of lead sulfide captured on the paper, measured by the integrated density of each well area (Supplementary Data 12). Quantification of H2S production was performed by measuring the integrated density using ImageJ, compared to a well next to it that contained no protein for background.

We used two approaches to monitor H2S levels (see below). With this strategy we detected H2S and not other volatile sulfur compounds. For example, methanethiol is a thiol while H2S is not. Although methanethiol and H2S are each volatile, they have distinct reactivities. The lead acetate assay detects volatile H2S, which generates lead sulfide (which is black). Although this assay has its limitations[52], it does not detect methanethiol and was used in the past to distinguish those two molecules in food sample analysis[64–66]. As for the MeRho-Az probe described in the next section, the mechanism of detection is based on a difference in chemical reactivity of H2S vs organic thiols, as tested by the authors who developed the probe[34].

**Detection of H2S levels by confocal microscopy.** For the quantification of H2S levels, worms were synchronized and grown at 20 °C on regular NGM plates seeded with OP50-1 until they reached late L4 stage. At this point, 50 animals per strain were transferred to fresh plates containing fluorescent H2S probe to develop until the next day. Plates with H2S sensor were made by spreading 100 μl of 40 μM MeRho-Az solution (in DMSO; MeRho-Az was synthesized in-house following a previously described protocol[34]) on the plate surfaces and left to dry for at least 4 h. On the control plates, the same volume of DMSO was spread as a vehicle control. On the day 1 of adulthood, worms were collected to a tube containing M9 buffer and centrifuged for 1 minute at 40 × g to remove bacteria. Fixation was done in 2% PFA for 20 min at 37 °C followed by incubation with 4% PFA (Sigma Aldrich, #P6148) for 20 min at RT with shaking. PFA was removed and worms were washed 3 times with PBS supplemented with 0.01% Triton and twice with PBS. PBS was removed and mounting media (Ibidi GmbH, Martinsried Germany, ref. 50001) was added directly to the tube. Worm suspension was transferred to the glass slide, covered with a cover slip and sealed with the nail polish. Samples were recorded on Leica TCS SP8 DLC Digital Light Sheet and Confocal microscope using ×10 air objective and VIS (488 nm) laser. Obtained images were first processed with Worm-align open source pipeline for straightening and then analyzed for fluorescence intensity using Cell Profiler software.

**Persulfidation detection by confocal microscopy.** Worms were synchronized by putting 15 gravid adults to lay eggs for 2 h. Once the animals reached day 1 of adulthood, they were washed off the plates with M9 buffer, collected into a 1.5 ml

tube, and centrifuged for 1 min at $400 \times g$. After 3 washes, M9 was removed, and worms were snap frozen in liquid nitrogen. Samples were defrosted by putting the tubes shortly in the water bath, and 200 µl of 5 mM NBF-Cl (Sigma Aldrich, #163260) in PBS supplemented with 0.01% Triton was immediately added to tubes, followed by incubation at 37 °C for 1 h with shaking. Worms were washed for 1 minute with ice-cold methanol while mixing, followed by 3 washes with PBS-Triton to remove excess NBF-Cl. Methanol/acetone fixation was performed on ice by incubating the samples in the ice-cold methanol for 5 min and then with the ice-cold acetone for 5 min. Acetone was removed, and 3 washes with PBS-Triton were performed. Samples were then again incubated with 5 mM NBF-Cl for 30 min at 37 °C to ensure complete labelling. After the washes were performed in the same order as previously described, worms were incubated with 150 µl of 25 µM DAz-2:Cy-5 click mix (mix was made in-house following a previously described synthetic protocol[35]) for 1 h at 37 °C with shaking. For the negative control, worms were incubated with 25 µM DAz-2:Cy-5 click mix prepared without DAz-2. Samples were then washed 2 times for 5 minutes with ice-cold methanol and 3 times for 5 minutes with PBS-Triton to remove excess of the preclick product. DAPI staining was performed by incubating the samples with 300 nM DAPI solution in PBS-Triton for 5 min at RT with agitation. After several washes with PBS-Triton and PBS, worms were mounted on glass slide. Samples were recorded on Leica TCS SP8 DLC Digital Light Sheet and Confocal microscope using ×10 air objective and 405 nm laser for DAPI, 488 nm laser for NBF-adducts and 635 nm laser for PSSH. Images were first processed with the Worm-align open source pipeline for straightening and then analyzed for fluorescence intensity using Cell Profiler software.

**Scoring of transgenic promoter-driven GFP.** For P*atf-4*(uORF)::GFP, L4 stage transgenic animals were exposed to chemicals by top-coating with 500 µl of each reagent (alpha-amanitin (Sigma #A2263), cycloheximide (Sigma #C7698), tunicamycin (Sigma #T7765), sodium arsenite (Honeywell International #35000)) or control (DMSO or M9 buffer) onto 6 cm NGM OP50 plates for 30 min to 4 hours, except that rapamycin (LC laboratories) was added to the NGM agar as described[6]. Then GFP fluorescent levels were either (1) scored or (2) quantified. (1) GFP scoring: Transgenic animals were first inspected with a dissecting scope while on still on the plate. GFP intensity was scored in the following categories: $0 =$ none or very low GFP usually corresponding to untreated control, $1 =$ low, $2 =$ medium, and $3 =$ high GFP fluorescence visible. Animals were washed off chemical treated plates, washed again at least twice, placed on OP50 NGM plates and were picked from there and mounted onto slides and GFP fluorescence was scored using a Zeiss AxioSKOP2 or a Tritech Research BX-51-F microscope with optimized triple-band filter-sets to distinguish autofluorescence from GFP at 40x as described[67]. GFP was scored as the following: None: no GFP (excluding spermatheca), low: either only anterior or only posterior of the animal with weak GFP induction, Medium: both anterior and posterior of the animal with GFP but no GFP in the middle of the animal. High: GFP throughout the animal. $P$ values were determined by Chi[2] test. (2) Quantification of GFP fluorescent levels: Animals were washed off reagent-containing plates, washed an additional two times, then placed into 24-well plates containing 0.06% tetramisole dissolved in M9 buffer to immobilize animals. Fluorescent pictures were taken with the same exposure settings (1 s) at ×10 magnification using an Olympus Cellsens Standard Camera on an inverted microscope. GFP levels were assessed by drawing a line around the animal, measuring mean grey value and using the same area next to it for background using ImageJ. The arbitrary fluorescent value corresponds to mean grey value of the animals minus the background.

**Western blot.** About 5000 *C. elegans* (L4 or day-1-adults indicated in figure legends) were sonicated in lysis buffer (RIPA buffer (ThermoFisher #89900), 20 mM sodium fluoride (Sigma #67414), 2 mM sodium orthovanadate (Sigma #450243), and protease inhibitor (Roche #04693116001)) and kept on ice for 15 min before being centrifuged for 10 min at $15,000 \times g$[68]. For equal loading, the protein concentration of the supernatant was determined with BioRad DC protein assay kit II (#5000116) and standard curve with Albumin (Pierce #23210). Samples were treated at 95 °C for 5 min, centrifuged for 1 min at $10,000 \times g$ and 40 µg protein was loaded onto NuPAGE Bis-Tris 10% Protein Gels (ThermoFisher #NP0301BOX), and proteins were transferred to nitrocellulose membranes (Sigma #GE10600002). Western blot analysis was performed under standard conditions with antibodies against Tubulin (1:500, Sigma #T9026), GFP (1:1'000, Roche #11814460001), Cystathionase/CTH (1:2000, abcam #ab151769), Puromycin (1:10,000, Millipore #MABE343), and Phospho-eIF2α (Ser51) (1:1000, Cell Signaling #9721). HRP-conjugated goat anti-mouse (1:2000, Cell Signaling #7076) and goat anti-rabbit (1:2000, Cell Signaling #7074) secondary antibodies were used to detect the proteins by enhanced chemiluminescence (Bio-Rad #1705061). For loading control (i.e., Tubulin) either corresponding samples were run in parallel, membrane was cut if the size of Tubulin and protein of interest were not overlapping, membrane was incubated with loading control after detection of protein of interest on the same blot, or the blot was stripped (indicated in figure legends). For stripping, membranes were incubated for 5 min in acid buffer (0.2 M Glycine, 0.5 M NaCl, pH set to 2 with HCl) and afterwards for 10 min in basic buffer (0.5 M Tris, pH set to 11 with NaOH) and washed with TBS-T before blocking. Quantification of protein levels was determined by densitometry using ImageJ software

and normalized to loading control (i.e., Tubulin). Uncropped blots are provided in the Source data File.

**Reporting summary**. Further information on research design is available in the Nature Research Reporting Summary linked to this article.

## Data availability

The RNA sequencing data in this publication have been deposited in NCBI's Gene Expression Omnibus and are accessible through GEO Series accession number GSE173799. Figure legends refer to the raw and source data. Source data are provided with this paper.

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

## Acknowledgements

We thank Alex Hofer, Anita Goyala, Sara Schütze, Carolin Imse, Julia Rogers, and Lorenza E. Moronetti Mazzeo for help with scoring lifespan, stress, and GFP assays, Michael Steinbaugh for help with the initial analysis of the RNA sequencing data, Stephanie Lin for contributing to earlier stages of this work, S. Mitani and the National BioResource Project for the *atf-4(tm4212)* and *tm4397)* alleles, Mike Crowder for the *rars-1(gc47)* allele, Chi Yun and David Ron for the P*atf-4*(uORF)::GFP reporter strain, WormBase for curated gene and phenotype information, Jay Mitchell and Nancy Pohl for comments on the manuscript, and Spalentor and Michael Hall for inspiration. Some strains were provided by the CGC, which is funded by the NIH Office of Research Infrastructure Programs (P40 OD010440). Portions of this research were conducted on the Orchestra High Performance Computer Cluster at Harvard Medical School (NCRR 1S10RR028832-01). Supported by funding from the Swiss National Science Foundation PBSKP3_140135, P300P3_154633, and PP00P3_163898 to C.Y.E. and C.S., and PZ00P3-185927 to A.L., the Leenaards and Novartis Foundation to A.L., ETH Research Grant (ETH-30-16-2) to R.V., the Iacocca Family Foundation to J.M., and the NIH to T.K.B. (R35 GM122610), along with a DRC grant from the NIDDK (P30 DK036836). MRF and DP were funded by the European Research Council (ERC) under the European Union's Horizon 2020 research and innovation programme (Grant Agreement No. 864921). Part of this research was conducted while Collin Y. Ewald was an Ellison Medical Foundation/AFAR Postdoctoral Fellow.

## Author contributions

C.Y.E. and T.K.B. conceived the study and designed the experiments. All authors participated in analysing and interpreting the data. C.Y.E., K.P., M.B., S.R.S., C.S., and R.V. performed lifespan assays. C.Y.E., M.B., C.S., and R.V. performed oxidative stress assays. C.Y.E., M.B., and C.S. performed thermotolerance assays. C.Y.E., M.B., K.P., S.R.S., and R.V. scored GFP reporters. C.S., R.E., J.M., A.L., and D.P. performed H2S measurement assays. P.L. and C.H. performed Ribo-sequencing analysis. D.P. and M.F. performed persulfidation assays and analysis. C.S and C.Y.E. analyzed transcription profiles. C.Y.E., C.S., and J.M. performed qRT-PCR. J.M. and R.V. performed the Western blots and puromycin assays. I.M. and W.B.M. generated transgenic strains. C.Y.E. performed all other assays. C.Y.E., T.K.B., and J. M. wrote the manuscript in consultation with the other authors. C.S. and J.M. contributed equally to this work. C. Y. E. and T.K.B. jointly supervised the work.

## Competing interests

The authors declare no competing interests.
