## [Peer Review File · Nature Communications]

ATF-4 and hydrogen sulfide signalling mediate longevity in response to inhibition of translation or mTORC1Editorial Note: This manuscript has been previously reviewed at another journal that is not operating a transparent peer review scheme. This document only contains reviewer comments and rebuttal letters for versions considered at Nature Communications.

REVIEWERS' COMMENTS

Reviewer #1 (Remarks to the Author):

The manuscript has considerably improved and is much more focused in the new revised version. My main concerns are addressed, this is an important and timely study. I fully support its publication in Nature Communications without delay.

I have one minor comment: The sentence in Line 59 "The effects of ATF4 on metazoan longevity have not been explored" should be edited given that there is a survival experiment in Fig 4 of Rousakis et al (PMID: 23692540)

Martin Denzel

Reviewer #3 (Remarks to the Author):

This is a revised manuscript, transferred from a sister journal. In this revised version, the authors have performed additional experiments to address the concerns reviewers raised and significantly edited the data included in this paper to be more focused on the mTORC1/translation/ATF-4/CTH-2/H2S axis of story. Most significantly, the new imaging-based assay for persulfidation provides stronger data supporting their claims.

Overall, the manuscript is greatly improved. However, it is rather regretful that a few questions remain unanswered in the current version of the manuscript. For example, while the author provides new dataset supporting the idea that mTORC1 inhibition exert its effect on ATF-4 via translation reduction, it acts through an eIF2a independent mechanism. Unfortunately, it remains unknown what this "eIF-2a independent mechanism" is. Similarly, it remains unknown why aft-4 is indispensable for mTORC1 or mTORC2- mediates lifespan but dispensable for lifespan phenotypes induced by mTORC1/C2 double mutants. Having this said, these unanswered questions may be addressed in their future studies. Finally, I feel that the remove of mouse data weakens the potential impacts of this paper.

Reviewer #4 (Remarks to the Author):

Response to reviewer comments/suggestions

Ewald, Blackwell, and colleagues have addressed the reviewers concerns/suggestions impressively. Therefore, I recommend accepting this manuscript for publication.

Notably, the ATF-4 signaling pathway presented in this manuscript is complex, and therefore there are still open questions (as acknowledged by the authors). However, I am positive that these questions will be addressed in future papers and inspire other research groups to follow these exciting observations and hypotheses.

Specific recommendations/comments

1. According to the Wormbase information, the T04C10.11 gene is part of the 5' UTR of atf-4. This gene appears to be affected by SIR-2.1. However, the authors did not discuss its function.

2. There are other volatile sulfur molecules, e.g., methyl mercaptan, that can be produced by cells. The authors should provide references indicating that the assays used to measure H₂S are indeed specific to H₂S or that such molecules are not synthesized in C. elegans. Otherwise, the authors should discuss the limitations of the H₂S assays, particularly the selectivity toward H₂S.

REVIEWERS' COMMENTS

Reviewer #1 (Remarks to the Author):

The manuscript has considerably improved and is much more focused in the new revised version. My main concerns are addressed, this is an important and timely study. I fully support its publication in Nature Communications without delay.

I have one minor comment: The sentence in Line 59 “The effects of ATF4 on metazoan longevity have not been explored” should be edited given that there is a survival experiment in Fig 4 of Rousakis et al (PMID: 23692540)

Response: We thank the reviewer for their positive comments and suggestion. As the reviewer requested, we have edited the sentence in Line 59 and the sentence is now “It remains to be determined whether ATF4 promotes metazoan longevity”. We have also edited Lines 111-113 to incorporate the results shown in this reference, which indicate that a partial loss of atf-4 function does not reduce wild type lifespan.

Reviewer #3 (Remarks to the Author):

This is a revised manuscript, transferred from a sister journal. In this revised version, the authors have performed additional experiments to address the concerns reviewers raised and significantly edited the data included in this paper to be more focused on the mTORC1/translation/ATF-4/CTH-2/H2S axis of story. Most significantly, the new imaging-based assay for persulfidation provides stronger data supporting their claims.

Overall, the manuscript is greatly improved. However, it is rather regretful that a few questions remain unanswered in the current version of the manuscript. For example, while the author provides new dataset supporting the idea that mTORC1 inhibition exert its effect on ATF-4 via translation reduction, it acts through an eIF2a independent mechanism. Unfortunately, it remains unknown what this “eIF-2a independent mechanism” is. Similarly, it remains unknown why aft-4 is indispensable for mTORC1 or mTORC2- mediates lifespan but dispensable for lifespan phenotypes induced by mTORC1/C2 double mutants. Having this said, these unanswered questions may be addressed in their future studies. Finally, I feel that the remove of mouse data weakens the potential impacts of this paper.

Response: We thank the reviewer for their positive comments. Our data support that reduced translation per se mediates the effect of mTORC1 inhibition on ATF-4, a claim that we make in the paper. We agree with the reviewer that these issues could be further studied in the future, and also that that is beyond the reasonable scope of the current manuscript.

Reviewer #4 (Remarks to the Author):

Response to reviewer comments/suggestions

Ewald, Blackwell, and colleagues have addressed the reviewers concerns/suggestions impressively. Therefore, I recommend accepting this manuscript for publication.

Notably, the ATF-4 signaling pathway presented in this manuscript is complex, and therefore there are still open questions (as acknowledged by the authors). However, I am positive that these questions will be addressed in future papers and inspire other research groups to follow these exciting observations and hypotheses.

Response: We thank the reviewer for their positive comments.

Specific recommendations/comments

1. According to the Wormbase information, the T04C10.11 gene is part of the 5' UTR of *atf-4*. This gene appears to be affected by SIR-2.1. However, the authors did not discuss its function.

Response: The reviewer is correct that the T04C10.11 gene is part of the 5' UTR of atf-4. According to the latest mapping information on Wormbase, the microarray probe SMD_T04C10.4 that was found affected by SIR-2.1 now points to both atf-4 and T04C10.11. Therefore, it is not clear whether T04C10.11 is actually regulated by SIR-2.1. In addition, considering the totality of evidence reported here, including the evolutionary conservation of ATF-4 regulation and functions, it is unlikely that this gene has a substantial effect on our results. We have added a few sentences to the legend of Extended data Fig. 1b in which we discuss the possibility that peptides encoded by small ORFs in the atf-4 5' UTR might have influenced our results.

2. There are other volatile sulfur molecules, *e.g.*, methyl mercaptan, that can be produced by cells. The authors should provide references indicating that the assays used to measure H_2S are indeed specific to H_2S or that such molecules are not synthesized in *C. elegans*. Otherwise, the authors should discuss the limitations of the H_2S assays, particularly the selectivity toward H_2S .

Response: Methanethiol is a thiol while H₂S is not. Although methanethiol and H₂S share volatility as a feature, they have quite different and distinct reactivity. We used two approaches to monitor H₂S levels. The lead acetate assay detects volatile H₂S, which would generate black lead sulfide. Although it has its limitations (Filipovic et al, Chem Rev, 2018), it does not react with methanethiol, and was used in the past to distinguish those two molecules in food sample analysis (Takai & Asami, Soil Sci & Plant Nutr, 1962; Dateo et al, J Food Sci, 1957; Farbood & MacNeil, J Food Sci, 1978). As for the MeRho-Az probe, the mechanism of detection is based on a difference in chemical reactivity

of H₂S vs organic thiols, as tested by the authors who developed the probe (Hammers et al, JACS, 2015). We have placed a note to this effect in the methods section (see H₂S production capacity assay).

References

Filipovic et al. *Chemical Biology of H₂S Signaling through Persulfidation*. *Chem Rev*, 2018, Feb 14;118(3):1253-1337. doi: 10.1021/acs.chemrev.7b00205.

Takai and Asami. *Formation of methyl mercaptan in paddy soils I*. *Soil Sci & Plant Nutr*, 1962, 8:3, 40-44. doi: 10.1080/00380768.1962.10430996.

Dateo et al., *Identification of the volatile sulfur components of cooked cabbage and the nature of the precursors in the fresh vegetable*. *Journal of Food Science*, 1957, doi: 10.1111/j.1365-2621.1957.tb17501.x.

Farbood and MacNeil. *Limitations of lead acetate for separation of methanethiol and hydrogen sulfide from food systems*. *J Food Sci*, 1978, doi: 43:139-140, 10.1111/j.1365-2621.1978.tb09753.x

Hammers et al. *A Bright Fluorescent Probe for H₂S Enables Analyte-Responsive, 3D Imaging in Live Zebrafish Using Light Sheet Fluorescence Microscopy*. *J. Am. Chem. Soc.* 2015, 137, 32, 10216–10223, doi: 10.1021/jacs.5b04196.